# GENERATIVE TOPOLOGY FOR SHAPE SYNTHESIS

## ABSTRACT

The *Euler Characteristic Transform* (ECT) is a powerful invariant for assessing geometrical and topological characteristics of a large variety of objects, including graphs and embedded simplicial complexes. Although the ECT is invertible in theory, no explicit algorithm for general data sets exists. In this paper, we address this lack and demonstrate that it is possible to *learn* the inversion, permitting us to develop a novel framework for shape generation tasks on point clouds. Our model exhibits high quality in reconstruction and generation tasks, affords efficient latent-space interpolation, and is orders of magnitude faster than existing methods.

## 1 INTRODUCTION

Understanding shapes requires understanding their geometrical and topological properties in tandem. Given the large variety of different representations of such data, ranging from point clouds over graphs to simplicial complexes, a general framework for handling such inputs is beneficial. The *Euler Characteristic Transform* (ECT) provides such a framework based on the idea of studying a shape from multiple directions—sampled from a sphere of appropriate dimensionality—and at multiple scales. In fact, the ECT is an injective map, serving as a *unique characterisation* of a shape (Ghrist et al., 2018; Turner et al., 2014). Somewhat surprisingly, this even holds when using a finite number of directions (Curry et al., 2022). Hence, while it is known that the ECT can be inverted, i.e. it is possible to reconstruct input data from an ECT, only algorithms for special cases such as planar graphs are currently known (Fasy et al., 2018). Hence, despite its advantageous properties (Dłotko, 2024; Munch, 2023), the ECT is commonly only used to provide a hand-crafted set of features for shape classification and regression tasks (Amézquita et al., 2021; Crawford et al., 2020; Marsh et al., 2024; Nadimpalli et al., 2023).

Recent work demonstrated that the ECT can be combined with deep-learning paradigms, leading to a fully-differentiable representation, which exhibits high computational and predictive performance in classifying shapes arising from point clouds or graphs (Röell & Rieck, 2024). This representation does not result in accessible latent space and therefore cannot be used to sample new ECTs or invert them. As one of the **contributions** of this paper, we overcome these restrictions and develop different deep-learning models for inverting the ECT when dealing with *point clouds*. Point clouds permit capturing objects in high resolution, while still providing a sparse representation of three-dimensional data. Their permutation-invariant nature makes them a challenging data modality for machine-learning algorithms, resulting in the development of highly-specialised architectures. For instance, Point-Voxel CNN (Liu et al., 2019), proposes an architecture that combines sparse voxel convolutions and an MLP acting on the point cloud directly, whereas DeepSets (Zaheer et al., 2017) develops a provably permutation-invariant network based on MLPs and suitable aggregation functions. However, representations of point clouds that are *intrinsically* permutation-invariant have the benefit that a wider range of machine-learning architectures become available, thus also permitting a diverse sets of tasks to be addressed. Our paper argues that the ECT is a representation with preferable properties, being (i) permutation-invariant by definition, and (ii) capable of learning rotations from the data. *We thus demonstrate for the first time how to efficiently use the ECT in generative tasks.* Specifically, we show that the ECT leads to generative models that outperform existing models both in terms of reconstruction/generation quality as well as in computational performance.

## 2 BACKGROUND AND METHODS

Point clouds are a ubiquitous data modality, often arising in the context of sensors, such as LiDAR in self-driving cars, or computer-aided design. They typically occur in large volumes, requiring ideally real-time inference with often limited computational resources. *Efficient and optimisable* architectures are therefore highly preferable for such applications. As we will subsequently demonstrate, the ECT considerably simplifies the architectural requirements, because its discretisation represents a point cloud as an image. Thus, a larger set of optimisation and compression techniques become available, making the ECT compatible with *generic* as opposed to *specialised* architectures.

### METRICS FOR POINT CLOUDS

Prior to discussing the ECT, we give a brief overview of metrics that we will use in the experimental section. These are motivated by the insight that the comparison of point clouds requires some form of (dis)similarity measure. A good metric should balance computational speed and theoretical guarantees; finding such metrics is a challenging task, since often computations require the consideration of all pairs of points between the two point clouds. Although not a metric in the mathematical sense, the *Chamfer Distance* (CD) poses a good balance between computational speed and quality. For point clouds $X$ and $Y$ it is defined as

$$\text{CD}(X, Y) = \sum_{x \in X} \min_{y \in Y} \|x - y\|_2^2 + \sum_{y \in Y} \min_{x \in X} \|x - y\|_2^2. \tag{1}$$

Work by Achlioptas et al. (2018) showed that CD-based losses result in reconstructions with *non-uniform* surface density, even for *uniformly-sampled* ground-truth data. Another common metric is the *Earth Mover's Distance* (EMD), which is based on concepts from optimal transport, i.e.

$$\text{EMD}(X, Y) = \min_{\phi: X \to Y} \sum_{x \in X} \|x - \phi(x)\|, \tag{2}$$

where $\phi$ refers to the image of $x$ under an optimal transport plan. Solving the optimal transport problem for the EMD is a computationally intensive task that can already become prohibitive for point clouds comprising a few thousand points.

### EULER CHARACTERISTIC TRANSFORMS

We first describe the Euler Characteristic Transform (ECT) in the more general setting of simplicial complex before giving an explanation of our performance improvements for handling point clouds. Simplicial complexes extend the dyadic relations of graphs to incorporate higher-order elements (simplices) such as triangles or tetrahedra. These complexes are a natural modality for modelling data; 3D meshes can be considered 2-dimensional simplicial complexes, for instance, with the 2-simplices given by the (triangular) faces. Simplicial complexes and their *invariants*, i.e. characteristic properties that remain unchanged under transformations like homeomorphisms, play an important role in computational topology. A key feature of these invariants is that they are an *intrinsic* property, meaning that they do not depend on a specific choice of coordinates. An important (combinatorial) invariant is the *Euler Characteristic* $\chi$, defined as the alternating sum of the number of simplices in each dimension, i.e. $\chi(K) := \sum_{d=0}^{D} (-1)^d |K_d|$, where $K_d$ denotes the set of $d$-simplices of a $D$-dimensional simplicial complex $K$. To extend the expressivity of this invariant, we need to provide it with geometrical and topological information about the input data. This requires vertex coordinates for $K$, so that $K \subset \mathbb{R}^n$, and a continuous function $f \colon \text{vert}(K) \to \mathbb{R}$ defined on the vertices $\text{vert}(K)$ of $K$. Given $t \in \mathbb{R}$, we now consider the pre-image of $f$ of the sublevel set $(\infty, t]$, denoted by $K_t := f^{-1}((\infty, t])$. This pre-image includes a $k$-simplex if all its vertices, i.e. all its 0-simplices, are included in the pre-image, hence it is a subcomplex of $K$. The function $f$, also referred to as a *filtration function*, permits us to calculate the Euler Characteristic $\chi(K_t)$ of each pre-image $K_t$, leading to the *Euler Characteristic Curve* (ECC) induced by $f$. The main insight of the *Euler Characteristic Transform* (Turner et al., 2014, ECT) is that it is possible to use a family of filtration functions, parametrised by a *direction* on a sphere $S^{n-1}$, to obtain a highly-expressive representation of the simplicial complex as a family of curves. With a sufficient number of directions, the ECT becomes an injective function mapping each point cloud to a *unique* summary (Curry et al., 2022). In our discretised setting (see below), we typically choose a large number of directions to counteract the loss of precision.

In the context of our work on point clouds, we employ a filtration function $f$ based on *hyperplanes*. Calculating the Euler Characteristic alongside this filtration, we obtain an invariant that provides an expressive statistic *with* favourable scalability properties, both in terms of size (number of points) and in terms of dimension (number of coordinates per point). Given a direction vector $\xi \in S^{n-1}$, the hyperplanes *normal* to $\xi$ define a filtration function of the form

$$f\colon S^{n-1} \times \mathbb{R}^n \to \mathbb{R}$$
$$(\xi, x) \mapsto \langle x, \xi \rangle, \tag{3}$$

where $\langle , \rangle$ denotes the standard Euclidean inner product. Moreover, we define the *height h* of a point $x$ in the point cloud to be the value $f_\xi(x) := f(\xi, x)$. The ECT is then defined as

$$\mathrm{ECT}\colon S^{n-1} \times \mathbb{R} \to \mathbb{Z}$$
$$(\xi, h) \mapsto \chi\left(f_\xi^{-1}\big((-\infty, h]\big)\right). \tag{4}$$

When working with point clouds, we are essentially dealing with 0-dimensional simplicial complexes, so Eq. (4) affords an explicit representation: For $X$ a point cloud and a fixed direction, the corresponding ECC effectively counts the number of points *above* a hyperplane of the form $\langle x, \xi \rangle = h$ along a direction vector $\xi \in S^{n-1}$ and height $h \in \mathbb{R}$. To see this, notice that the sublevel set filtrations of $X$ with respect to the hyperplane are given by $X_h = \{x \in X | \langle x, \xi \rangle \leq h\}$ and by definition of the Euler Characteristic, we have $\chi(X_h) = |X_h|$, the cardinality of the set. A point $x \in X$ is included in $X_h$, thus affecting $\chi(X_h)$, if and only if its height $h_x = \langle x, \xi \rangle$ along $\xi$ is less than $h$. We can thus formulate the value of the ECT at a point $x$ in terms of an *indicator function*:

$$\mathbb{1}_x(\xi, h) := \begin{cases} 1 & \text{if } \langle \xi, x \rangle \leq h \\ 0 & \text{otherwise} \end{cases} \tag{5}$$

This permits us to write Eq. (4) as

$$(\xi, h) \mapsto \sum_{x \in X} \mathbb{1}_x(\xi, h). \tag{6}$$

Interchanging the indicator function for a smooth *sigmoid function* makes this discrete construction differentiable with respect to the direction $\xi$ as well as an input coordinate $x$, enabling its use as a machine-learning layer. Moreover, the sigmoid approximation also makes computations parallelizable, resulting in high throughput (Nadimpalli et al., 2023; Röell & Rieck, 2024).

In practice, the structure of the ECT permits us to represent its discretised version as an *image*, with rows in the image indexing the individual values in the filtration function, and the columns indexing the selected directions. Machine-learning models, operating on such image data, assume that neighbourhoods in the image are related and apply convolutions to process and parse features for downstream tasks such as classification or segmentation. To apply these models to the ECT, it is thus necessary that directions that are close together on the unit sphere are also close together in the image representation. In two dimensions, a single angle (parametrising the unit circle) is sufficient. However, in higher dimensions, there is no canonical parametrisation. While it is possible to parametrise the unit sphere with spherical coordinates and stack the resulting representation for each direction into a voxel grid, the memory and compute requirements scale *cubically* with the ECT's resolution, making the approach not scalable. We propose a different approach that embeds a unit circle along each pair of axes in the ambient space. For each circle we sample the directions along a regular interval to obtain a multi-channel image of the object. The number of channels in the image scales quadratically with the input dimension $n$ of the point cloud, since we consider each pair of axes for a total of $n(n-1)/2$ channels, thus posing only a slight limitation for extremely high dimensions. The main advantage of our approach is that we can use CNNs, which are well-suited for multi-channel images. We find that this representation provides sufficient expressivity to encode equivariance with respect to orientation through data augmentation, which we will further explore in the experiments.

Our last improvement to the ECT concerns its *invertibility*. Being an injective mapping, the pre-image of an ECT is guaranteed to be unique. Nevertheless, to this date, there are no known generally-applicable procedures for inverting the ECT. Our differentiable approximation of the ECT permits us to use machine-learning models to *learn* the inversion, provided sufficient training data are available. We thus formulate the inversion as training an encoder–decoder model. The encoder turns input point clouds into an ECT, whereas the decoder aims to *reconstruct* a point cloud from an ECT. We realise both of these steps using an MLP. Figure 1 provides a high-level overview of our pipeline.

Figure 1: Given a point cloud on the left, we compute its *Euler Characteristic Transform* (ECT), which results in a compressed representation. For generative tasks, we train a generative model (middle) to reconstruct and generate the distribution of shapes. The (possibly-generated) ECT is then passed through the encoder model to obtain the reconstructed point cloud. Our pipeline is decoupled, permitting *any* generative image model to be used to generate point clouds. Further image compression can be employed to obtain a highly-compact representation of the input data.

## TOPOLOGICAL LOSS FUNCTIONS

Although the EMD has been used as a loss term for various point-cloud processing tasks, the computational requirements limit its practicality. Next to the improved ECT calculations, we thus propose a novel *topologically-inspired loss term* that is *both* density-aware and efficient to compute. Tracking the Euler Characteristic along $h$ we obtain the cumulative histogram of $X$ along a given direction $\xi$. From this cumulative histogram, we can approximate the density through the derivative with respect to $h$: If the ECT along each direction is approximated with a smooth sigmoid function and we calculate its derivative, we obtain a density estimate. In essence, we obtain a *directional kernel density estimate* with the kernel equal to the derivative of the sigmoid function. A kernel density estimate centres a kernel function, often a Gaussian, around each data point and estimates the density through the summation of centred kernel functions. In our case, the points are the heights and around each height, we centre the derivative of the sigmoid function, which resembles a Gaussian, while approximating the density through the summation. Mathematically, this results in

$$
\mathrm{DECT}\colon S^{n-1} \times \mathbb{R} \to \mathbb{R},
$$
$$
(\xi, h) \mapsto \sum_{x \in X} S'(h - \langle x, \xi \rangle), \tag{7}
$$

where $S'$ is the derivative of the sigmoid function. We may further normalise these density estimates along each direction $\xi$ to obtain a 'directional' probability density function. Given two such density estimates for point clouds $X$ and $Y$, we obtain a measure of how well the densities along a direction align by computing the KL-divergence. Fixing a finite number of directions, this leads to

$$
D_{\mathrm{T}}(X, Y) := \sum_{\xi \in \Xi} D_{\mathrm{KL}}(\mathrm{DECT}_X(\xi, h), \mathrm{DECT}_Y(\xi, h)). \tag{8}
$$

Being density-aware and computationally efficient, Eq. (8) results in a suitable term for regularising a CD-based loss, thus constituting a fast and viable alternative to losses based on the EMD.

## 3 EXPERIMENTS

Having a theoretical pipeline for reconstructing point clouds in various dimensions in place, we perform comprehensive experiments to understand both qualitative and quantitative properties of our methods. Our experiments comprise three parts:

  (i) We first evaluate reconstruction and generation performance on a benchmark dataset.
 (ii) We then show that we learn *equivariant representations* without requiring architectural changes.
(iii) Finally, we show that interpolating between ECTs leads to smooth transitions between shapes.

**Architectures.** We use two ECT-based architectures, an ECT-MLP that encodes an ECT into a point cloud and model based on *variational autoencoders*, denoted ECT-VAE, that can both generate and reconstruct ECTs. With the latter model, we thus obtain a pipeline for generating novel points or reconstructing them from a latent representation. Our ECT-MLP model consists of a standard MLP architecture with 4 layers and ReLU activation functions. For 2D data we use 512 hidden neurons

Table 1: Reconstruction results on the three ShapeNetCore15k classes. To simplify comparisons, the CD is scaled by $10^4$ and the EMD is scaled by $10^3$. ECT-MLP denotes a model trained on the original data, whereas ECT-MLP-N is trained on the normalized data. To obtain a fair comparison, we evaluate both types of models (original and normalised) on both versions of the dataset. Please refer to Appendix A for a more detailed performance comparison.

| | Airplane | | Chair | | Car | |
|---|---|---|---|---|---|---|
| Model | CD ($\downarrow$) | EMD ($\downarrow$) | CD ($\downarrow$) | EMD ($\downarrow$) | CD ($\downarrow$) | EMD ($\downarrow$) |
| **Original dataset** | | | | | | |
| PointFlow | 1.30 ± 0.00 | 5.36 ± 0.06 | *6.94 ± 0.01* | **10.43 ± 0.02** | 17.54 ± 0.16 | 12.93 ± 0.19 |
| SoftFlow | 1.19 ± 0.00 | *4.28 ± 0.06* | 11.05 ± 0.03 | 17.68 ± 0.08 | 6.82 ± 0.01 | 11.44 ± 0.10 |
| ShapeGF | **1.05 ± 0.00** | 4.42 ± 0.04 | **5.96 ± 0.01** | *12.23 ± 0.11* | **5.68 ± 0.01** | *9.26 ± 0.18* |
| ECT-VAE (Ours) | 1.67 ± 0.01 | 5.00 ± 0.09 | 15.96 ± 0.07 | 18.47 ± 0.17 | 10.27 ± 0.06 | 12.01 ± 0.27 |
| ECT-MLP (Ours) | 1.32 ± 0.00 | 4.85 ± 0.08 | 14.78 ± 0.04 | 18.30 ± 0.11 | 7.27 ± 0.01 | 10.76 ± 0.17 |
| ECT-MLP-N (Ours) | *1.16 ± 0.00* | **3.30 ± 0.04** | 10.43 ± 0.02 | 13.22 ± 0.09 | *6.36 ± 0.01* | **7.68 ± 0.12** |
| **Normalised dataset** | | | | | | |
| PointFlow | 8.68 ± 0.02 | 35.19 ± 0.72 | 42.93 ± 0.07 | 70.55 ± 0.43 | 38.96 ± 0.47 | 66.99 ± 0.72 |
| SoftFlow | 7.93 ± 0.01 | *28.14 ± 0.35* | 45.27 ± 0.10 | 71.19 ± 0.18 | 38.99 ± 0.74 | 60.02 ± 0.75 |
| ShapeGF | **7.02 ± 0.01** | 29.33 ± 0.33 | **24.44 ± 0.07** | **48.30 ± 0.38** | **27.15 ± 0.07** | *43.70 ± 0.83* |
| ECT-VAE (Ours) | 11.07 ± 0.11 | 32.81 ± 0.59 | 65.51 ± 0.28 | 73.56 ± 0.51 | 163.24 ± 7.46 | 75.53 ± 1.71 |
| ECT-MLP (Ours) | 8.83 ± 0.03 | 31.63 ± 0.48 | 60.64 ± 0.15 | 73.06 ± 0.39 | 61.29 ± 0.50 | 57.52 ± 0.65 |
| ECT-MLP-N (Ours) | *7.69 ± 0.03* | **21.72 ± 0.53** | *42.72 ± 0.10* | *52.95 ± 0.42* | *30.48 ± 0.05* | **36.10 ± 0.48** |

per layer, while for 3D data, we increase this to 2048. Our reasoning is that the output dimension increases significantly, from $2 \times 512 = 1024$ in the 2D case to $3 \times 2048 = 6168$ in the 3D case. Our ECT-VAE model is based on a convolutional VAEs (Higgins et al., 2016). Its encoder consists of 5 convolutional layers followed by a linear embedding layer to a 256-dimensional latent space. The number of channels for each convolutional layer are 32, 64, 128, 256, and 512, respectively, with the encoder following these channel sizes in reverse. We hypothesise that more elaborate architectures, such as diffusion models or vision transformers could yield even better results in terms of quality while lacking computational efficiency.

**Experimental Setup and Evaluation.** We train our ECT-MLP and ECT-MLP-N models with a loss based on the Chamfer Distance. Additionally, we add our topological loss term to serve as an additional regularisation term. By contrast, we train ECT-VAE with using a loss based on the KL-divergence and the MSE between the original ECT and its reconstruction. All models use the Adam Optimizer with a learning rate of $1.00 \times 10^{-3}$ for the 2D datasets and $5.00 \times 10^{-4}$ for the 3D datasets. For all datasets except ShapeNetCore15k we train models using all available classes, thus learning both inter- and intra-class distributions. To evaluate reconstructions, we follow the setup of Yang et al. (2019), which reports the maximum mean discrepancy (Gretton et al., 2012, MMD) based on the Chamfer Distance (MMD-CD) or the Earth Mover's Distance (MMD-EMD) between reconstructed point clouds. To evaluate generative performance, we use the 1-nearest-neighbour accuracy metric (1-NNA), which measures the accuracy in distinguishing between the original dataset or the generated dataset using a 1-NN classifier trained on the CD or the EMD metric. While common practice, this metric is limited in that it requires the input distribution to be sufficiently diverse in terms of geometrical properties or rotations.

## 3.1 Reconstructing and Generating Shapes

Our first set of experiments assesses the reconstruction and generation capabilities of our methods, arguably the most important part of a new model, using a subset of the ShapeNetCore15k benchmark dataset (Yang et al., 2019). Following common practice, the dataset consists of 2048 samples of three shape classes (airplane, chair, and car). Objects in the dataset are neither centred with respect to the origin nor are they scaled uniformly; in fact, the radius of their bounding sphere is normally distributed. Subsequently, we refer to this dataset as the *original* ShapeNetCore15k dataset, noting that the objects are generally not centred and their bounding box does not have unit radius. We also provide a *normalised* version of this dataset, in which we centre each object with respect to its barycentre and axial mean, and rescale its bounding sphere to have unit radius. This will enable us to focus on comparing reconstruction and generative qualities without accounting for size intra-

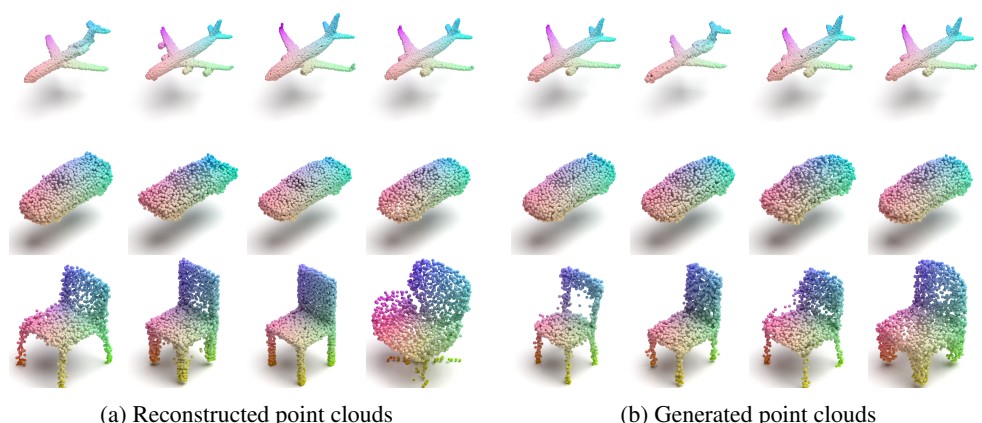

|     |     |
| --- | --- |
| (a) Reconstructed point clouds | (b) Generated point clouds |

Figure 2: Examples of *reconstructed* (left, using ECT-MLP) and *generated* (right, using ECT-VAE) point clouds for the three classes in the ShapeNetCore15k dataset.

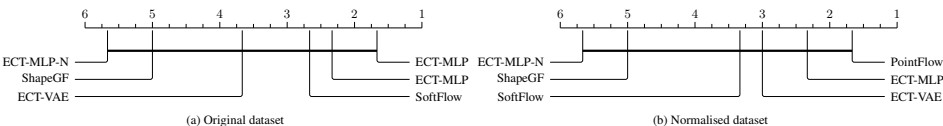

Figure 3: Critical difference plots of the reconstruction performance (in terms of the Chamfer Distance) of all models. Differences in reconstruction performance are not statistically significant.

class size differences. Following the literature, we report the mean Chamfer Distance (CD) and the Earth Mover's Distance (EMD), calculated with respect to reconstructions on the validation dataset and averaged across 10 runs to capture stochasticity. We use several state-of-the-art models as **comparison partners**, namely (i) PointFlow (Yang et al., 2019), (ii) SoftFlow (Kim et al., 2020), (iii) ShapeGF (Cai et al., 2020), and (iv) SetVAE (Kim et al., 2021).[1]

Table 1 depicts the results of the *reconstruction* task, while Figure 2 depicts several example point clouds. We first observe that our methods are *consistently* among the top three methods on the original dataset and among the top two methods on the normalised dataset. Notably, on the original dataset, our ECT-MLP-N, which is trained on the normalised dataset but evaluated[2] on the original dataset leads to the best reconstruction performance of all ECT-based models. This indicates that individual differences in the radius of bounding spheres serve as a confounding factor in assessing reconstruction performance. Our assessment on the normalised dataset corroborates this; here, our ECT-MLP-N model exhibits performance on a par with much more complicated models like ShapeGF.

However, while these results provide some measure of how models behave across different runs of the same experiment, they do not permit a direct insight into the variance of reconstruction quality within the *test* dataset since we found that all comparison partners report performance only on the validation split of the input data. We follow this incorrect practice here in the main text to make our results comparable; Appendix A presents a detailed comparison on the test dataset, in which our method outperforms *all* existing methods in terms of quality. Summarising these additional experiments, we observe that differences in reconstruction performance of all models can be explained by a small number of 'outlier point clouds,' which result in high variance. This suggests that all models (including ours) are performing similarly for the most part. We substantiate this claim by calculating *critical difference plots* (Demšar, 2006), which assesses to what extent differences in reconstruction performance are statistically significant. Figure 3 depicts critical difference plots for the CD metric. A horizontal line connects models whose performance differences are not statistically significantly different. As we can see, in our case, no model is statistically significantly better or

---

[1]Due to its architecture, SetVAE cannot encode and reconstruct an input point cloud directly, which is why we only assess its performance in a *point cloud generation task*.

[2]Please see below for additional details on the evaluation procedure.

Table 2: Generative results on the three classes of ShapeNetCore15k. We report the 1-NNA for the MMD-CD and MMD-EMD for each of the three classes and highlight the winner per column in bold text, with the second place being shown in italics. We observe that there is strong variability between all comparison partners, and no model clearly outperforms all others. Our models perform on a par with all comparison partners.

| | Airplane | | Chair | | Car | |
|---|---|---|---|---|---|---|
| Model | CD (↓) | EMD (↓) | CD (↓) | EMD (↓) | CD (↓) | EMD (↓) |
| **Original dataset** | | | | | | |
| PointFlow | 75.68 | **69.44** | 60.88 | **59.89** | *60.65* | 62.36 |
| SoftFlow | **70.92** | **69.44** | *59.95* | 63.51 | 62.63 | 64.71 |
| ShapeGF | 80.00 | *76.17* | 68.96 | 65.48 | 63.20 | **56.53** |
| SetVAE | *75.31* | 77.65 | **58.76** | *61.48* | **59.66** | 61.48 |
| ECT-VAE (Ours) | 76.05 | 77.90 | 60.37 | 73.72 | 62.46 | 73.04 |
| ECT-VAE-N (Ours) | 87.53 | 79.26 | 68.05 | 68.20 | 62.36 | *58.81* |
| **Normalised dataset** | | | | | | |
| PointFlow | *61.48* | 71.48 | *61.48* | **58.23** | *61.48* | **52.84** |
| SoftFlow | 78.89 | **68.83** | 64.80 | 67.22 | 67.61 | 58.80 |
| ShapeGF | 80.62 | 83.46 | **60.42** | 60.80 | **59.23** | *55.40* |
| SetVAE | 88.89 | 79.14 | 64.05 | 64.05 | 68.75 | 63.49 |
| ECT-VAE (Ours) | 81.48 | 88.64 | 65.91 | 82.24 | 69.86 | 77.04 |
| ECT-VAE-N (Ours) | **56.67** | 79.88 | 62.36 | *58.81* | 62.64 | 68.20 |

worse than any other (the same results hold for the EMD). Hence, all things being equal, in practice, the selection of a model should be foremost dictated by *computational performance*, i.e. by the training and inference (generation) time.

Table 3: Inference time (T) in ms, measured on GPU or CPU, and number of parameters (P) in millions for each model.

| Model | Device | T(ms) | P (M) |
|---|---|---|---|
| PointFlow | | 270.00 | 1.60 |
| SoftFlow | | 120.00 | 31.00 |
| ShapeGF | GPU | 340.00 | 4.80 |
| SetVAE | | 30.00 | 0.75 |
| ECT-VAE (Ours) | | 0.79 | 46.00 |
| ECT-VAE (Ours) | CPU | 5.88 | 46.00 |

To assess this aspect, Table 2 depicts the results of the *generation* task. Here, we are restricted to our VAE-based model ECT-VAE, since it is the only one that directly exposes a latent space for sampling followed by reconstruction. Similar to the reconstruction task, we find that our models typically is in the top three performers, exhibiting high generative performance across all the datasets. Interestingly, the generative performance of ShapeGF, arguably the best model in the reconstruction task, is markedly lower, underscoring once again our observation that there is no clear 'best' model in terms of reconstruction and generation performance. Notably, as Table 3 demonstrates, our model outperforms all comparison partners by multiple orders of magnitude in terms of *inference time*. This even holds in case we use the CPU for generating point clouds, with inference times still being about 6 times faster than the fastest GPU-based comparison method. This makes our method suitable for high-quality and high-performance point cloud processing even in settings where no GPU is available. We envision that further optimizations, such as pruning, quantising, and compiling the model to a suitable format will further reduce inference times. Next to the fast inference time, our model also exhibits fast training time, along the order of SetVAE, requiring about 5–7 hours in total. This highlights the fact that the ECT combined with a conceptually simple model can easily perform on a par with more involved architectures.

Despite this advantageous properties, our experiments also uncovered two drawbacks of ECT-based models. The first being the comparatively large number of parameters in the model, of which most reside in the first and last layer. While we believe that the conceptual simplicity of our model potentially permits reducing the final number of parameters (a task we aim to tackle in future work),

Figure 4: Our ECT-MLP model can capture *equivariance* with respect to rotations through data augmentation. The orientation of reconstructed point clouds (top) and original point clouds (bottom) is matched perfectly on the MNIST dataset.

we are still the 'largest' model. A second drawback is that the ECT is inherently susceptible to scale differences. If an input dataset exhibits large differences in terms of the size of bounding spheres, care needs to be taken such that the ECT has sufficient capacity to pick up details at all resolutions. As we observed, scale differences can lead to large relative errors (but small absolute errors) during point cloud reconstruction, prompting us to assess our method on a normalised version of the data. In the future, we want to make ECT-based models intrinsically aware of shape differences, for instance using an additional layer for scaling and translating the point cloud. A limitation concerning all models is the fact that existing evaluation metrics are not adequately capturing generative quality. Subsequently, we sidestep this issue by analysing generated point clouds with known geometrical-topological characteristics.

We end this section with a brief discussion of post-processing steps that are required to train and evaluate our models in mixed scenarios, such as the ones shown in Table 1. For instance, we may train a model on the normalised dataset, disregarding all scale information, but evaluate it on the original dataset. To accomplish this, we store the mean position and scale of objects of a given class and rescale the point cloud created by our model. When assessing reconstruction performance, we believe that the scale and spatial position of a point cloud should not matter. Our results on the normalised dataset indicate that spatial position and scale serve as confounding factors for model performance in the sense that a small translation or rescaling of the generated point cloud carries large penalties.

### 3.2 LEARNING EQUIVARIANT REPRESENTATIONS

Table 4: Reconstruction results for an equivariant learning task on the Manifolds and MNIST datasets. As a baseline, we report CD between the dataset and a random rotation of the samples.

| | **Manifolds** | |
| --- | --- | --- |
| | ECT-MLP | Random Rotation |
| Sphere | 61.32 ± 4.51 | 80.48 ± 7.43 |
| Torus | 56.34 ± 8.91 | 1525.10 ± 671.84 |
| Cube | 75.77 ± 24.89 | 312.91 ± 83.22 |
| Möbius strip | 41.33 ± 27.89 | 4087.36 ± 2114.60 |
| | **MNIST** | |
| | 53.64 ± 14.98 | 635.34 ± 701.58 |

Recent work showed that *equivariance* with respect to certain operations like rotations can also be achieved through data augmentation, thus obviating the need for more complex architectures (Abramson et al., 2024). To assess the capabilities of our ECT-based models in this context, we follow Qi et al. (2017) and use a point-cloud version the MNIST dataset of handwritten digits. During training, we apply a random rotation to each point cloud and then compute the ECT, thus permitting the model to learn an equivariant representation of the data. Notice that for such 2D data, rotations correspond to a cyclic column permutation of the ECT. As Figure 4 shows, data augmentation is sufficient to encode rotations, resulting in an equivariant model without having to specifically add equivariance as a separate inductive bias. Motivated by these promising results, we repeat the experiment in three dimension with a novel synthetic dataset consisting of point clouds sampled from 2-manifolds, i.e. spheres, tori, cubes, and Möbius strips. Each object is randomly rotated and the task is to reconstruct both the right type of manifold and orientation of the object. As opposed to the 2D case, learning $SO(3)$-equivariance from the data is in general a challenging task in point cloud processing, which typically requires specialised architectures that contain equivariance biases. However, as in the 2D example, we observe that even though ECT-MLP is not $SO(3)$-equivariant by design, the model nevertheless learns to decode the orientation from the ECT. Table 4 depicts the results for both datasets, proving that our model not only learned to reconstruct the *right* object but also learned its orientation. Learning equivariance through data augmentation as opposed to specific architectural changes poses another advantage of our ECT-based models.

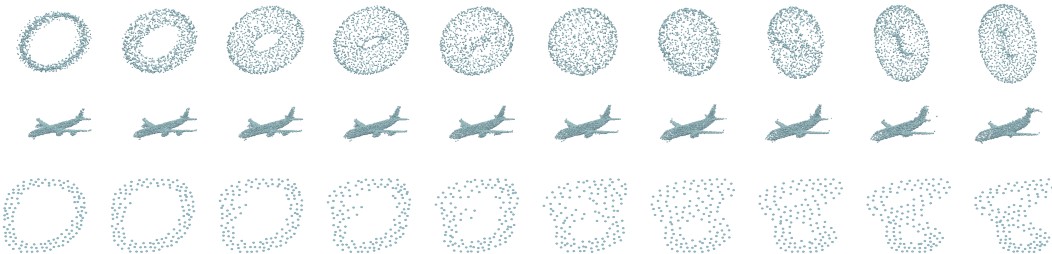

Figure 5: Given two ECTs, we apply a linear interpolation between them and encode each intermediary step into a point cloud on three different datasets (top: manifolds, middle: 'airplanes' class of ShapeNetCore15k, bottom: MNIST). Although it is not guaranteed that each intermediate ECT is the image of a valid shape, we observe that the encoder still reconstructs geometrically plausible point clouds. We thus do not have to specifically constrain the latent space to obtain suitable reconstruction, which is beneficial for general point cloud processing. For the 'manifolds' dataset, it is remarkable that the orientation of the Möbius strip (source point cloud) and the orientation of the torus (target point cloud) are very different. This implies that the latent space also encodes orientations.

### 3.3 INTERPOLATING BETWEEN SHAPES

As our final experiment, we consider each ECT to be an element of a (disentangled) latent space, and we *interpolate* between the ECT of different classes on the manifolds dataset (from the previous section), the 'airplane' class of ShapeNetCore15k, and MNIST. This is important to understand the characteristic properties of the ECT since the ECT, while *injective* on the space of all shapes, fails to be *surjective*, i.e. not every ECT is a plausible representation of a shape. We remark that our representation of the ECT as an image enables us to efficiently interpolate on a per-pixel basis. To assess the quality of the latent space, we reconstruct each ECT during the interpolation using ECT-MLP. Figure 5 depicts the resulting point clouds. Latent representations remain 'plausible,' even when interpolating between manifolds like a Möbius strip and a torus, whose topological characteristics differ substantially. In this case, we find that the ECT-MLP first changes the Möbius strip into a sphere, which is subsequently changes into a torus. This process entails changing the orientation of the encoded object, meaning that the ECT encodes information about the orientation in the latent space. This leads us to conclude that the ECT results in advantageous latent space for point cloud processing, since it permits us to control the orientation of the objects before, during, and after the interpolation.

## 4 CONCLUSION AND DISCUSSION

In this paper, we develop the first approach to *efficiently invert* a geometrical-topological descriptor, the Euler Characteristic Transform, and show its efficacy in reconstructing and generating shapes. Our pipeline captures characteristic properties of different datasets and uses the ECT as an *intrinsic* and integral part of the model. We also propose an extension for *synthesising* new shapes by sampling the corresponding latent spaces. Despite its simplicity, our model produces high-quality and diverse results that are on a par with or even exceed the reconstruction and generation quality of methods with more involved architectures. Our model is orders of magnitude faster than existing methods, thus permitting real-time shape generation on both the CPU and the GPU. Moreover, our experiments show that both the correct shape *and* the orientation are learned from the data, leading to an intrinsic approximation of equivariance. Finally, we explored the use of the ECT as a latent space, finding high-quality intermediate reconstructions and smooth interpolations between both their shapes and reconstructions. For future work, we aim to explore (i) directly imbuing ECT-based schemes with equivariance properties, (ii) developing novel latent-space interpolation schemes based on optimal transport, and (iii) extending our methods to graphs and simplicial complexes.

## REPRODUCIBILITY STATEMENT

We will provide the code and configurations for our experiments to ensure reproducibility. All experiments were run on a single GPU to enable the comparison of results.

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

Table 5: The reconstruction results for the Airplane, Chair and Car dataset reported over the *test* set. Chamfer Distance is multiplied by $10^4$ and the EMD is multiplied by $10^3$. The literature reports results for reconstruction over the *validation* set and for consistency that practise is followed in the main text. In the table below we report the standard deviation of the loss of the individual reconstructed samples wheras the main text reports the standard deviation of the mean. We report the results over both the original and normalised dataset and report the filtered, by Interquartile Range, results indicated with (IQR). The large standard deviation suggests that the quality of the reconstructions can significantly differ from object to object.

| | Airplane | | Chair | | Car | |
|---|---|---|---|---|---|---|
| | CD ($\downarrow$) | EMD ($\downarrow$) | CD ($\downarrow$) | EMD ($\downarrow$) | CD ($\downarrow$) | EMD ($\downarrow$) |
| **Original dataset** | | | | | | |
| PointFlow | 5.12 ± 5.73 | 12.99 ± 13.50 | 9.73 ± 38.79 | 14.32 ± 12.83 | *10.32 ± 10.70* | 18.25 ± 12.22 |
| ShapeGF | 3.64 ± 4.26 | 8.91 ± 6.51 | *8.01 ± 29.60* | *10.16 ± 8.22* | **6.13 ± 4.12** | **12.51 ± 7.42** |
| SoftFlow | 6.48 ± 8.81 | 12.28 ± 11.36 | 9.84 ± 35.33 | 12.74 ± 10.30 | 11.35 ± 13.14 | 17.83 ± 11.43 |
| ECT-MLP | 6.65 ± 6.83 | 11.03 ± 7.36 | 10.09 ± 38.32 | 11.87 ± 9.41 | 14.49 ± 14.22 | 19.61 ± 13.59 |
| ECT-MLP-N | **1.16 ± 0.64** | **3.28 ± 1.40** | **6.37 ± 2.04** | **7.60 ± 3.56** | 10.41 ± 10.07 | *13.09 ± 9.44* |
| ECT-VAE | *1.67 ± 1.37* | *5.09 ± 2.41* | 10.17 ± 3.96 | 11.49 ± 5.09 | 16.01 ± 14.39 | 18.42 ± 12.60 |
| **Original dataset (IQR)** | | | | | | |
| PointFlow | 3.92 ± 2.76 | 10.09 ± 6.66 | 6.94 ± 1.89 | 12.84 ± 6.81 | *8.98 ± 4.11* | 16.48 ± 8.27 |
| ShapeGF | 2.65 ± 1.67 | 7.79 ± 4.26 | **6.15 ± 1.62** | *9.11 ± 4.14* | **5.62 ± 2.43** | **11.08 ± 4.38** |
| SoftFlow | 4.48 ± 3.60 | 10.35 ± 7.01 | 7.22 ± 2.04 | 11.32 ± 5.34 | 9.47 ± 4.78 | 16.00 ± 7.62 |
| ECT-MLP | 5.13 ± 3.65 | 9.53 ± 4.36 | 7.45 ± 2.42 | 10.96 ± 5.14 | 12.34 ± 6.19 | 17.21 ± 7.97 |
| ECT-MLP-N | **1.04 ± 0.18** | **3.10 ± 1.09** | *6.23 ± 1.45* | **7.26 ± 3.01** | 9.19 ± 4.65 | *11.45 ± 4.70* |
| ECT-VAE | *1.33 ± 0.34* | *4.85 ± 1.91* | 9.60 ± 2.77 | 10.93 ± 4.25 | 13.73 ± 7.24 | 15.95 ± 6.91 |
| **Normalised dataset** | | | | | | |
| PointFlow | 25.93 ± 26.02 | 65.62 ± 59.15 | 53.99 ± 205.25 | 69.39 ± 73.21 | 42.69 ± 38.09 | 74.66 ± 48.78 |
| ShapeGF | 18.54 ± 20.36 | 45.25 ± 29.57 | *36.31 ± 107.32* | *47.46 ± 34.09* | **25.54 ± 18.20** | **50.20 ± 29.94** |
| SoftFlow | 32.29 ± 38.73 | 61.88 ± 52.82 | 52.68 ± 190.48 | 62.31 ± 59.18 | 46.76 ± 47.26 | 72.71 ± 45.93 |
| ECT-MLP | 33.80 ± 32.09 | 56.56 ± 33.75 | 137.58 ± 2379.66 | 67.31 ± 311.14 | 59.92 ± 56.03 | 78.79 ± 53.75 |
| ECT-MLP-N | **7.68 ± 4.16** | **21.78 ± 9.21** | **30.49 ± 9.20** | **35.70 ± 15.28** | *42.61 ± 46.26* | *52.96 ± 39.71* |
| ECT-VAE | *11.02 ± 8.73* | *32.92 ± 15.20* | 171.13 ± 1712.56 | 75.27 ± 271.10 | 65.02 ± 61.55 | 73.19 ± 50.24 |
| **Normalised dataset (IQR)** | | | | | | |
| PointFlow | 20.82 ± 14.40 | 53.90 ± 34.02 | 32.72 ± 9.02 | 60.41 ± 32.77 | *37.13 ± 19.39* | 68.12 ± 35.18 |
| ShapeGF | 14.05 ± 8.31 | 40.49 ± 20.43 | **28.48 ± 7.53** | *43.28 ± 19.61* | **23.12 ± 11.30** | **44.39 ± 18.86** |
| SoftFlow | 22.92 ± 17.64 | 53.33 ± 34.50 | 33.90 ± 9.88 | 53.52 ± 25.13 | 39.15 ± 21.99 | 65.04 ± 31.90 |
| ECT-MLP | 27.78 ± 20.11 | 49.38 ± 19.51 | 35.06 ± 11.07 | 51.05 ± 22.11 | 51.24 ± 28.67 | 70.25 ± 33.99 |
| ECT-MLP-N | **6.95 ± 1.37** | **21.25 ± 8.31** | *29.09 ± 6.85* | **33.90 ± 12.25** | 37.43 ± 20.91 | *46.24 ± 21.40* |
| ECT-VAE | *9.07 ± 2.50* | *31.32 ± 11.99* | 46.43 ± 13.76 | 52.35 ± 20.36 | 55.28 ± 31.87 | 63.89 ± 28.42 |

# A ADDITIONAL ANALYSES ON RECONSTRUCTION PERFORMANCE

We provide additional and complementary analysis of the reconstruction experiment. The distribution of the reconstruction loss within the testset provides valuable insight on the variance of the quality. Low variance implies similar performance of the model accross all elements in the testset, whereas high variance implies considerable variance in quality. In addition to the tables in the main text, we re-evaluate the models on the testset and vizualise the results in Figure 6a. The results suggests that outliers have a large influence on the mean, further exacerbated by the assymetric nature of the loss term. To provide further insight into the distribution of the loss, we eliminate outliers with respect to the Interquartile Range (IQR) and present the results in (Figure 6b). Even with the outliers removed, variance remains large compared to the differences in mean between the models, suggesting large variance in reconstruction quality amongst *all* models. Table 5 shows the reconstruction results both with and without outliers.

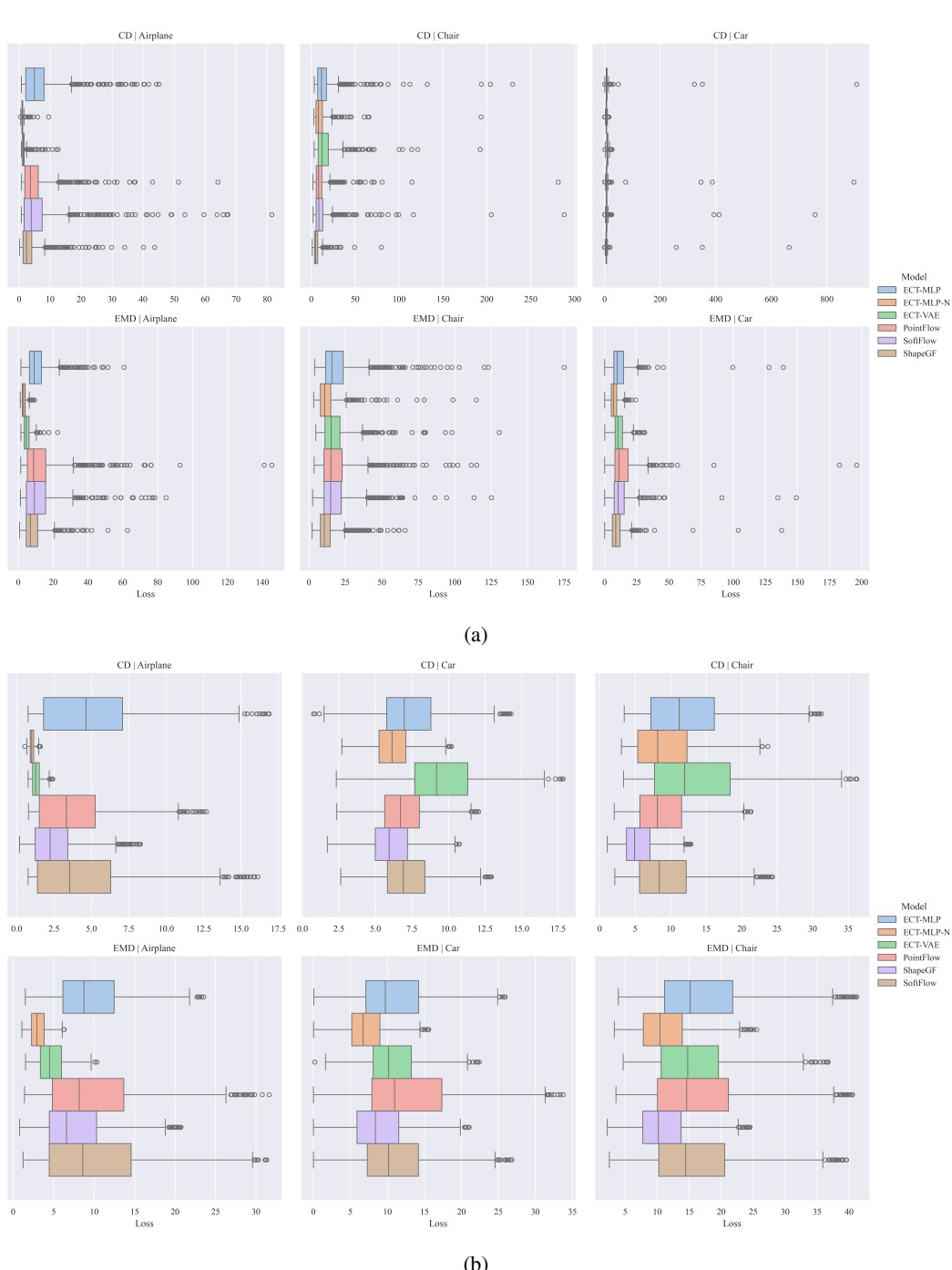

Figure 6: A visual representation of the EMD and CD loss per model and category, reported over the *test set* of the original data. The CD is multiplied by $10^4$ and EMD is multiplied by $10^3$. Figure 6a shows the results and for extra comparison, while Figure 6b shows the results without outliers, based on the Interquartile Range. The outliers, particularly for the category of cars, have a major influence on the reported mean of the loss, potentially skewing the results. With outliers removed, standard deviations remain large compared to the differences in the mean.

