# OpenReview forum: "Generative Topology for Shape Synthesis"
_ICLR.cc/2025/Conference — ICLR 2025 Conference Withdrawn Submission_

### Official Review · Reviewer_S7sm · 2024-10-31

**Soundness:** 1
**Presentation:** 2
**Contribution:** 2
**Rating:** 3
**Confidence:** 5

**Summary:**

This work proposes a framework for point cloud generation that employs the Euler Characteristic Transform (ECT) to obtain a representation. An additional encoder is then learned to invert the ECT and produce a point cloud, allowing a generative model (VAE) to be applied to the ECT representation. Experimental results show that the proposed framework achieves comparable performance to some baseline methods with higher time efficiency. Additionally, the framework attempts to learn an approximately equivariant representation.

**Strengths:**

- The exploration of employing ECT in 3D point cloud generation appears novel. It also demonstrates the possibility of learning an equivariant representation in the 2D case (Figure 4).
- The proposed framework achieves faster inference times compared to various baselines.

**Weaknesses:**

However, this work has several critical issues:

- Limited Novelty:
    - As mentioned in the introduction, the application of ECT on 3D point cloud analysis has been well explored in (Röell & Rieck, 2024). The formulation of ECT is directly adopted from (Röell & Rieck, 2024) without modification, and the writing and description of the formulation are also highly similar to (Röell & Rieck, 2024).
    - The main contribution, introducing an additional encoder to invert ECT representation into an explicit point cloud via a point cloud generative network, is considered trivial.
- Evaluation Performance:
    - While the authors claim that the proposed framework achieves comparable performance, several recent baselines are omitted, including:
        - [1] LION: Latent Point Diffusion Models for 3D Shape Generation. NeurIPS 2022.
        - [2] Autoregressive 3D Shape Generation via Canonical Mapping. ECCV 2022.
        - [3] Shape Generation and Completion Through Point-Voxel Diffusion. ICCV 2021.
        - [4] Diffusion Probabilistic Models for 3D Point Cloud Generation. CVPR 2021.
    - The authors argue that existing processing networks, such as Point-Voxel networks, are unsuitable for point cloud representation but provide no comparison. Notably, [1] and [4] adopt PVCNN for point cloud generation, and a comparison should be provided to support this claim.
    - Moreover, [2] and [3] learn a latent representation that can faithfully reconstruct point clouds. Before claiming the proposed representation's superiority, reconstruction performance and efficiency must be compared with these works.
- Non-equivariant Network:
    - Despite showing some effectiveness (Table 4), the current network (ECT-MLP) is not inherently equivariant, which weakens the discussion on learning equivariant representations.

Justification of Rating:
This work explores the use of ECT for point cloud generation tasks. However, its technical contribution is limited, primarily based on a direct adoption of the formulation in (Röell & Rieck, 2024) without significant modifications. The comparisons with existing methods lack several important baselines to support its claims. Despite being an interesting direction, the manuscript requires substantial revision to reconsider its formulation and include additional experiments. Therefore, I recommend rejecting the paper and encouraging the authors to resubmit after addressing these issues.

**Questions:**

- Design of VAE:
    - The current approach only explores VAE as the generative model. Given that various generative models could apply to the proposed framework, why is VAE only considered in this case?

**Details Of Ethics Concerns:**

N.A.

---

### Official Review · Reviewer_Uh9G · 2024-11-03

**Soundness:** 3
**Presentation:** 3
**Contribution:** 2
**Rating:** 6
**Confidence:** 4

**Summary:**

The paper presents a framework for shape generation tasks on point clouds using the Euler Characteristic Transform (ECT). The authors demonstrate that the ECT can be inverted, allowing for the development of a generative model in reconstruction and generation tasks. The model is shown to be efficient in terms of latent-space interpolation and computational speed, outperforming existing methods by a significant margin. The paper also introduces a topologically-inspired loss function that is both density-aware and efficient to compute, and explores the use of the ECT as a latent space for shape interpolation.

**Strengths:**

1. The inversion of the Euler Characteristic Transform for shape generation is an innovative approach. The paper provides a thorough explanation of the ECT and its application in generative tasks, backed by solid theoretical foundations.
2. The model's efficiency in reconstruction and generation tasks, especially in terms of computational speed, is a significant advantage over existing methods. Comprehensive experiments and comparisons with state-of-the-art models are provided, demonstrating the effectiveness of the proposed approach.
3. The paper shows that the ECT can be used for a variety of tasks, including reconstruction, generation, and learning equivariant representations.

**Weaknesses:**

1. While the model is efficient, it has a comparatively large number of parameters, which could be a drawback in some applications. Further optimization to reduce the number of parameters without compromising performance could be beneficial.
2. The paper notes that the ECT is susceptible to scale differences in the input dataset. Addressing this issue to make the model more robust would be an improvement.
3. More experiments on diverse datasets could further demonstrate the generalization capabilities of the model, not only on three categories (chair, airplane, car). More complex shape with various topologies should be evaluated to demonstrate the core idea of the proposed method.
4. While the model performs well, providing more insight into how the ECT encodes and decodes shape information could help readers better understand the underlying mechanisms. Figure 1 is to transfer the coarse idea to readers.
5. The performance does not achieve the best from the quantitative evaluations; some point cloud generation on diffusion model should be considered as the baseline to compare.
[1] Lion: Latent point diffusion models for 3d shape generation
6. The limitations and failure cases should be demonstrated in the paper.

**Questions:**

The paper is well-written and presents a well contribution to the field of shape generation. Based on the above observations, I am interested at the new tools to develop the shape generation, but there evaluation is not sufficient only on three small categories, and the performance is not the best over the other baselines, and discussion on the failure cases. Currently, I lean towards positive for looking forward to the responses.
Detailed questions please refer to weaknesses.

---

### Official Review · Reviewer_9vtt · 2024-11-03

**Soundness:** 2
**Presentation:** 2
**Contribution:** 2
**Rating:** 3
**Confidence:** 4

**Summary:**

The paper extend an existing technique - Euler Characteristic Transform - which uses  scalar functions assigned to vertices of a simplicial complex in order to produce a signature that is guaranteed to be invertible. The authors propose to extend this specifically to point cloud, and show that the invertibility of the ECT can enable 3D reconstruction and generation of point clouds.

**Strengths:**

The ECT is a great idea as a method to represent geometry within machine learning, and I believe it can lead to great additional discoveries. Additionally, the focus on such topolgical concepts in the context of point clouds (which do not hold any inherent topology except the trivial one) is a good idea.

**Weaknesses:**

I am a bit confused by the proposal by the authors. While the ECT seems like a powerful approach, its application to point clouds is a bit confounding, considering point clouds do not hold any topological information beyond that of a set, and the Euler Characteristic reduces to a trivial formula for them. Indeed, the attempt to discuss point clouds as simplicial complexes seems somewhat forced, albeit mathematically correct. As far as I understand, without having an actual underlying topology implied the the simplicial complex structure, the power of ECT is significantly reduced, and as far as I understand, the method reduces to classifying points based on their "depth" with respect to selected "view directions" - this does not strike me as extremely novel nor powerful. The results do not seem to show significant improvement over the quite saturated SotA.
 I additionally note that in several places, the authors are not mathematically rigorous, e.g.,
"A key feature of these invariants is that they are an *intrinsic* property meaning that they do not depend on a specific choice of coordinate", followed by "To extend the expressivity of this invariant, we need to provide it with geometrical and topological information about the input data" in which the authors proceed the embed the SC in 3D coordinates and exactly violate the notion of an intrinsic property they focus on before.

**Questions:**

- can you explain what is the exact power you see in using ECT, considering a point cloud has no underlying topological information and the EC becomes trivial?
- What is the exact difference between using ECT, and using the "z axis" in various arbitrary Cartesian 3D coordinate system (i.e., choose a random coordinate system, choose z as "f")?

---

### Official Review · Reviewer_fKuL · 2024-11-04

**Soundness:** 2
**Presentation:** 1
**Contribution:** 2
**Rating:** 3
**Confidence:** 4

**Summary:**

The paper proposes a novel point cloud generation framework which incorporates Euler Characteristic Transform (ECT). ECT is an injective mapping that is naturally permutation invariant. By converting point clouds into ECT maps and treating them as latent representations, a simple, non-permutation-invariant VAE can be used to generate the ECT maps. Converting ECT maps back to point clouds is difficult, thus the paper trained an MLP for the task using Chamfer loss in addition to a differentiable ECT-based loss, proposed in Röell & Rieck, 2024.
The proposed model is compared to existing works on ShapeNet15K and MNIST point cloud datasets. the model achieves a performance close to previous works, while inferencing much faster.

**Strengths:**

* The paper is the first in applying ECT to a generative task i.e. point cloud generation.
* The paper illustrates the possibility of solving the difficult, ECT-inversion problem using a simple MLP.
* The proposed method is at least an order of magnitude faster than previous works, possibly due to a the use of a simpler neural network.

**Weaknesses:**

* Over-generalization. The paper attempts to stay general about the dimension of the point clouds, but he writing is not always consistent. For example, in the paragraph starting at L138, "spherical coordinate", "voxel grid", and "scale cubically" are only appropriate for 3D scenario. However, what comes after implies arbitrary dimensions. This creates confusions. It might be a good idea to stay consistent and put 3D-specific and general contents to different sections / paragraphs.
* There needs major improvement on the writing. For example, there are factual errors such as L161 "We realise both of these steps using an MLP", which is not the case as the encoding process is done with ECT algorithm. There are also inconsistencies such as the purpose of the "encoder" in Figure 1 not matching the text in L161, and broken sentences such as L178 "Next to the improved ECT calculations, we thus...".
* L146-147 Cubic scaling of memory and compute w.r.t. the resolution of polar coordinate ECT: This is likely not true. The compute and memory will scale linearly with the number of angles used in the ECT, regardless of the dimension. Could you elaborate how you have arrived at the conclusion?
* The performance of the proposed method is not very good, lagging behind the previous works in almost all the settings.
* The rotation augmentation experiment (Figure 4) does not add any value, as any method can acquire equivariancy through data augmentation, and there lacks any comparisons with other methods.
* Reproducibility: The resolution of the ECT map is not known. This also creates a confusion on the expected input shape of ECT-MLP.
* The permutation invariant property of ECT is not fully taken advantage of. The decoder MLP converting the ECT to points is not permutation equivariant, possibly hindering the quality.

**Questions:**

* How would the resolution of the ECT affect the quality?
* Would other means of parameterizing the ECT angle work? E.g. a spiral on a unit sphere.
* Is it possible or useful to replace the ECT-to-point cloud network with a non-parametric method such as gradient descent using differentiable ECT (Röell & Rieck, 2024)?

---

### Note · Authors · 2024-11-18

**Comment:**

Dear reviewers,

We first would like to thank the reviewers for their extended reviews and interesting insights. It is encouraging to see that the core intent of the paper has been recognized!

Our aim has been to show that the ECT is a strong way to represent geometry in ML and a *first* attempt to inverting topological descriptors. To the best of our knowledge: No such method—neither non-parametric nor machine learning based—exists to date although some attempts have been made in rather specialized cases. Reflecting on your reviews, we find that this point requires _more_ emphasis, that inverse problems in topology are rather hard and non-trivial. The attempts so far, focusing on the Persistent Homology Transform, for instance, are able under favorable (!) circumstances to invert small planar graphs with up to roughly 50 nodes. For the ECT, by contrast, no such method exists, not even for point clouds and therefore it is remarkable that our straightforward architecture was able to achieve the results it did.

We are thus of the opinion that our work carries intrinsic value to the research community. At the same time, following your feedback, we feel that a substantially revised manuscript with an extended experiment section and more contextualization of the problem at hand would require another round of reviews, which is beyond the scope of this conference. Therefore, we decided to *withdraw* the submission and focus on addressing your concerns in a more thorough way than a rebuttal may provide us with.

We again would like to thank you all for your extensive and actionable feedback—we hope that you will get the chance to review a revised, improved version of this manuscript!

**Withdrawal Confirmation:**

I have read and agree with the venue's withdrawal policy on behalf of myself and my co-authors.